# Impact of CytoSorb Hemoadsorption Therapy on Fluid Balance in Patients with Septic Shock

**DOI:** 10.3390/jcm13010294

**Published:** 2024-01-04

**Authors:** Klaus Kogelmann, Tobias Hübner, Matthias Drüner, Dominik Jarczak

**Affiliations:** 1Department of Anesthesiology and Intensive Care Medicine, Klinikum Leer, Augustenstraße 35-37, 26789 Leer, Germany; 2Department of Anesthesiology and Intensive Care, Kantonsspital Münsterlingen, Spitalcampus 1, 8596 Münsterlingen, Switzerland; tobias.huebner@stgag.ch; 3Department of Anesthesiology and Intensive Care Medicine, Klinikum Emden, 26721 Emden, Germany; m.druener@klinikum-emden.de; 4Department of Intensive Care Medicine, University Medical Center Hamburg-Eppendorf, 20246 Hamburg, Germany; d.jarczak@uke.de

**Keywords:** inflammation, septic shock, CytoSorb, hemoadsorption, fluid balance, vasopressors, vascular barrier function, endothelial integrity

## Abstract

Recent in vitro studies have investigated the effects of hemoadsorption therapy on endothelial function in sepsis showing a reduction in markers of endothelial dysfunction, but, to this day, there are no clinical studies proving whether this approach could actually positively influence the disturbed vascular barrier function in septic conditions. We retrospectively analyzed data on administered fluid volumes and catecholamines in 124 septic shock patients. We collected catecholamine and volume requirements and calculated the volume balance within different time periods to obtain an assumption on the stability of the vascular barrier. Regarding the entire study cohort, our findings revealed a significant reduction in fluid balance at 72 h (T_72_) compared to both baseline (T_0_) and the 24 h mark (T_24_). Fluid balances from T_72_–T_0_ were significantly lower in hospital survivors compared with non-survivors. Patients who received a second catecholamine had a significantly lower in-hospital mortality. Our findings suggest that the applied treatment regimen including hemoadsorption therapy is associated with a reduced positive fluid balance paralleled by reductions in vasopressor needs, suggesting a potential positive effect on endothelial integrity. These results, derived from a large cohort of patients, provide valuable insights on the multiple effects of hemoadsorption treatment in septic shock patients.

## 1. Background

Sepsis represents a life-threatening condition that occurs when an infection exceeds local tissue containment and induces a chain reaction of dysregulated physiological responses that result in organ dysfunction [1,2]. Despite all efforts, mortality remains high, and sepsis is responsible for approximately 11 million deaths annually worldwide [3,4]. The endothelium, together with its most exposed component, the glycokalyx, act as sensors of infection and inflammation and are among the first line of immunological defenses against invading pathogens. They are responsible for maintaining normal vascular function and homeostasis, and are involved in various physiological processes such as angiogenesis, vascular tone regulation, and inflammation [5,6,7,8,9,10,11,12].

The early stages of sepsis are characterized by the activation of the innate immune system, triggering the release of inflammatory mediators, which activate a plethora of downstream cascades including the expression of various adhesion molecules, and also the coagulation system [11,12,13,14]. When combined, these mechanisms can ultimately lead to a proinflammatory and prothrombotic state, further contributing to tissue hypoxia and hypoperfusion, microvascular thrombosis, increased vascular permeability, capillary leakage, and overall impaired endothelial function, and ultimately organ dysfunction [12,15]. Although the Surviving Sepsis Campaign guidelines [16] emphasize the current insufficiency of evidence to provide a definitive recommendation on the utilization of blood purification techniques necessitating ongoing research in this field, there is promising data from individual technologies that could contribute to a more comprehensive understanding of their potential role in improving patient outcomes.

The CytoSorb adsorber is a hemoperfusion device based on a unique porous polymer bead technology which is capable of removing inflammatory mediators but also other predominantly hydrophobic substances (e.g., myoglobin, bilirubin) in a size range up to 60 kDa from whole blood in a concentration-dependent manner [17,18]. CytoSorb therapy has been shown to be safe and well tolerated among various indications and patient populations, with over 221,000 single treatments performed to date.

Recent in vitro studies suggest that hemoadsorption therapy may improve endothelial function in sepsis by reducing the levels of inflammatory mediators that contribute to endothelial activation and dysfunction [19,20]. Although there is a lot of basic research on the effects of hemoadsorption on endothelial function, there are no clinical studies on whether this approach could actually positively influence the disturbed vascular barrier function under septic conditions. Hypothetically, and in cases of a positive impact, this would clinically correlate with an improved fluid balance, less volume required for hemodynamic stabilization, and reduced catecholamine requirements. In our recently published study, we created a dynamic scoring system which allows for the assessment of hemodynamic status and development in the early phase of septic shock, enabling detection of a refractory status in septic patients and, consequently, differentiating them into subgroups with different mortalities [21]. We were able to show that the earlier hemoadsorption therapy was started, the better the outcome in terms of mortality.

Given this context, we analyzed the data on administered fluid volumes and catecholamines in the same cohort of patients in order to determine the potential benefit of CytoSorb hemoadsorption therapy on fluid balance and fluid requirements in a clinical scenario, in order to draw potential conclusions on the impact on endothelial function.

## 2. Material and Methods

### 2.1. Ethics Approval, Legal Considerations

The research received authorization from the ethics committee of the General Medical Council of Lower Saxony (reference number Bo/29/2019) and was conducted following the guidelines of the Declaration of Helsinki. Additionally, it adhered to the Good Clinical Practice Protocol (GCP) (2001/20/EEC, CPMP/ICH/135/95), established standard operating procedures, and the relevant laws and regulations of each respective country. The requirement for written informed patient consent was waived due to the retrospective nature of the study and the use of anonymized data collected during routine clinical care.

### 2.2. Study Design

Based on the collected data, we performed a retrospective analysis of 124 septic shock patients who had undergone adjunctive treatment with hemoadsorption therapy. The analysis incorporated data from three interdisciplinary intensive care units (ICUs) that followed similar procedures (Emden/Germany, Münsterlingen/Switzerland, UKE Hamburg/Germany). Inclusion criteria encompassed patients with a coded diagnosis of septic shock (in accordance with Sepsis-3 criteria). The definition of septic shock aligned with the Society of Critical Care Medicine (SCCM) and the European Society of Intensive Care Medicine (ESICM) sepsis-3 criteria [1], i.e., vasopressor requirement to maintain a mean arterial pressure (MAP) of 65 mmHg and serum lactate level >2 mmol/L in the absence of hypovolemia. Excluded were patients where data records were unavailable for analysis, patients not treated in the ICU and patients where norepinephrine (NE) requirement or lactate were not documented. Overall, treatment in our study strictly adhered to Surviving Sepsis Campaign Guidelines [16], encompassing timely source control and appropriate antibiotic therapy initiation.

### 2.3. Objectives

Fluid balance, administered fluid volumes, and catecholamine demand in regard to treatment time with hemoadsorption for the first 72 h were defined as primary objectives. ICU and hospital mortality, different laboratory values, e.g., lactate, inflammatory parameters (procalcitonin—PCT, C-reactive protein—CRP), creatinine, ventilator days, ICU, and hospital length of stay in regard to these balances were defined as secondary objectives.

### 2.4. Assessed Parameters

The following parameters were evaluated: medical history, patient characteristics, disease severity scores (Acute Physiology And Chronic Health [APACHE II], Simplified Acute Physiology Score [SAPS 2]), hemodynamics (catecholamine demand, heart rate, blood pressure), laboratory parameters (lactate clearance, inflammatory parameters, creatinine), initial volume requirement to achieve a MAP of 65 mmHg, use of either hydrocortisone or a second catecholamine (or both), CytoSorbtherapy-specific data (therapy delay after diagnosis of septic shock and start of standard therapy), duration of organ support (duration of mechanical ventilation, hemoadsorption therapy), outcome data (ICU and hospital stay and survival), and safety-relevant issues (adverse events). The amount of blood purified (ABP = duration of treatment * blood flow/body weight) was calculated as well, according to Schultz et al. [22].

### 2.5. Data Collection

Information was stored in anonymized tabular format. The consolidation and processing of the data was performed at the Department of Anesthesiology and Intensive Care at Emden Hospital, Germany.

### 2.6. Procedure

Following initial resuscitation, standard hemodynamic monitoring was consistently applied. However, in select cases, individualized monitoring strategies were implemented. This included the use of arterial lines, intermittent utilization of advanced hemodynamic monitoring such as PiCCO, and the occasional integration of ultrasound for a comprehensive assessment of hemodynamic status. We collected volume requirements for the first 72 h of hemoadsorption treatment. Administered volume boli to achieve a MAP of 65 mmHg were collected, as well as diuresis and other relevant parameters at each time period (T_0_ = time of volume administration at CytoSorb initiation, T_24_ = first 24 h of hemoadsorption treatment, T_72_ = 72 h after initiation of hemoadsorption treatment). We calculated the development in volume balance within these time periods to obtain an assumption on the stability of the vascular barrier in the dynamic process during early septic shock. We also calculated differences in catecholamine requirements to exclude the influence of catecholamine administration on fluid balance development. In addition, volume balances were calculated in relation to survival and in relation to the administration of a second catecholamine. We decided to restrict these data to the first 72 h, as this time period corresponds to the common treatment time with CytoSorb adsorbers in the literature.

This idea corresponds to the assumption that stabilization of the endothelial matrix implies a decreased permeability for fluids, so fluid requirements, and therefore the observed volume balance, should be reduced within the 72 h observational period.

### 2.7. Statistics

All primary and secondary variables were first assessed using an exploratory data analysis method and recorded descriptively. Data are reported as mean ± standard deviation or median as required. A normal distribution was tested using the Shapiro–Wilk test. Differences in primary endpoints between study populations were analyzed by two-way ANOVA and Wilcoxon test, *t*-test, Mann–Whitney U-Test, or χ2 test, as required. Correlations were tested with Spearman’s rho test. Data were analyzed with SPSS 20.0 (Armonk, NY, USA, IBM Corp.), and a value of *p* < 0.05 was defined as α (statistically significant).

## 3. Results

A total of 124 patients were included in the study; 37.9% (*n* = 47) of them were female. Hospital survival was 33.1% (*n* = 41), ICU survival was 37.9% (*n* = 47). In 43 patients, a second catecholamine was used (34.7%), while in 75 patients, hydrocortisone was applied (60.5%). Baseline characteristics are depicted in Table 1. Diagnoses/sources of infection in the study population included pneumonia being the predominant cause in the overall patient population (41.9%, *n* = 52). Among survivors, pneumonia was identified in 30.8%, while non-survivors presented a higher prevalence, at 69.2%. Abdominal infections were the second most common (41.1%, *n* = 51), with non-survivors (64.7%) outnumbering survivors (35.3%). As such, both patient cohorts (pneumonia and abdominal source) exhibited an almost equal distribution among survivors and non-survivors, with no significant differences observed (chi-square test, *p* ≥ 0.05). Urinary tract infections were present in 6.5% (*n* = 8) of patients (survivors 25%, non-survivors 75%), and miscellaneous sources in 13 patients (10.5%; survivors 38.5%, non-survivors 61.5%). The administration of hydrocortisone did not show a significant difference (chi-square test, *p* ≥ 0.05) between groups (of the *n* = 41 survivors, 68.3% received hydrocortisone, compared with 56.6% in the non-survivor group). All CytoSorb-treated patients (*n* = 124) also received concomitant continuous renal replacement therapy (CRRT) due to acute renal failure. Baseline characteristics are shown in Table 1.

In assessing variations in fluid balance across the entire patient cohort, a significant reduction was observed at day three (T_72_) in comparison with both the baseline (T_0_) and the 24 h mark (T_24_) (*p* < 0.001 for both) as depicted in Figure 1A,B.

The administered volume and fluid balance at time T_0_ demonstrated no significant differences in relation to mortality, with survivors receiving 76.2 mL/kg and non-survivors 82.5 mL/kg (*p* = 0.694) for volume, and 54 mL/kg for survivors and 55.26 mL/kg for non-survivors (*p* = 0.433) in fluid balance. However, at T_72_, administered volumes and fluid balances were significantly lower in survivors compared with non-survivors (Table 1). In the overall study population, we found a clear positive correlation between fluid balance and difference in catecholamine requirements (R = 0.26), i.e., the improvement in the volume balance was not influenced or bought at the cost of the administration of additional catecholamines. On the other hand, fluid balance showed a weak negative correlation with the amount of blood volume treated (R = −0.28, ABP in L/kg).

Fluid balances from T_72_–T_0_ were significantly lower in hospital survivors (median reduction 43.4 mL/kg) compared to hospital non-survivors (reduction 10.8 mL/kg, *t*-test *p* = 0.008, Cohen’s d = 0.596 shows a medium effect).

The association between fluid balance at 72 h and hospital mortality in patients not receiving a second catecholamine showed a significant difference between hospital survivors (*n* = 19, fluid balance reduction 45.4 mL/kg) and non-survivors (n=37, fluid balance reduction 8.5 mL/kg) (*p* = 0.008) (Figure 2 left and right).

There was a strong correlation between survival and necessity of a second catecholamine. Hospital survival probability was almost twice as high in patients who received a second catecholamine (odds ratio OR = 2.35 with 95% CI = [1.10; 5.03]) compared with those who did not (chi-square test, *p* = 0.02), i.e., sicker patients benefitted significantly from hemoadsorption therapy (SAPS 2 higher with 2nd catecholamine YES vs. NO: 59.6 vs. 53.4 *p* = 0.03 with 15% higher predicted mortality).

## 4. Discussion

In this study, we conducted a retrospective analysis of 124 septic shock patients who underwent CytoSorb hemoadsorption therapy to investigate its effect on fluid balance, which might serve as an indicator of endothelial stability and clinical outcomes. The primary objective was to assess fluid balance and catecholamine requirements during the first 72 h of treatment.

With regard to the entire study cohort, our findings revealed a significant reduction in fluid balance at 72 h (T_72_) compared with both baseline (T_0_) and the 24 h mark (T_24_), paralleled by a reduction in catecholamine needs. The administered volume and fluid balance showed no significant differences in relation to mortality at the onset of septic shock. However, after 72 h, a significant reduction in these volumes was observed in the group of patients who ultimately survived.

This decrease in volume administration and fluid balance as the key observation suggests an association and potential important positive clinical effect of the applied treatment regimen, including hemoadsorption therapy in regard to reduction of capillary leakage and vascular permeability.

In a recent experimental study, Jansen et al. [18] confirmed the “proof of principle” for CytoSorb hemoadsorption, showing effective attenuation of circulating cytokine levels during systemic hyperinflammation. Recent investigations in line with previous studies also provide data that the adsorber is capabable of adsorbing endothelial damaging proteins [19,20], suggesting positive effects on endothelial integrity. Knowing the central role of the endothelium in regulating various aspects of homeostasis, and knowing that hyperinflammatory conditions including septic shock lead to endothelial dysfunction resulting in microcirculatory and finally organ failure [9], there seems to be a sound rationale to support endothelial function and integrity by the removal of damaging substances. To date, however, only a few case reports have reported on positive effects on the fluid balance or extravascular lung water, respectively, where the patient is also receiving hemoadsorption therapy [23,24], so our data provide the first structured analysis of the effects of hemoadsorption in this regard. Effects regarding protection of the vascular barrier function, as also suggested in a case report by David et al. [25], would obviously present a completely new therapeutic approach in the field of sepsis management, in principle reducing the need for purely symptomatic fluid replacement.

Our finding that the fluid balance is significantly lower in hospital survivors is in line with various other studies reporting that a less positive fluid balance is associated with an improved outcome. Neyra and colleagues showed, in a total of 2632 patients, that higher cumulative fluid balance was independently associated with hospital mortality. These data refer to renal failure patients, but all our patients with hemoadsorption therapy in our evaluation suffered from acute renal failure (ARF) [26]. Similarly, in a recent systematic review and meta-analysis involving over 31,000 patients in 15 studies, Tigabu and colleagues found that a high fluid balance in the first 24 h to ICU admission increased the risk of death by 70% in patients with septic shock [27]. In contrast to this are data from Cronjohrt et al., who investigated the relationship between fluid balance and mortality in patients with septic shock in a post hoc analysis of the TRISS (Transfusions in Septic Shock) trial, didn’t find any statistically significant association between fluid balance and 90-day mortality; however, the study design had limited power for strong conclusions [28].

When considering fluid balance with correlation analysis, we found that lower reductions, which might be indicative of a less effective response to hemoadsorption but also caused by various other factors in the clinical course of the patient, were associated with almost constant norepinephrine requirements and a higher treated blood volume, suggesting that, despite fluid resuscitation and hemoadsorption therapy, vasopressor demands remained worse. In this regard, Lewejohanns and colleagues described the importance of appropriate fluid loading prior to the use of high catecholamine doses and on the influence of catecholamine demand [29]. This essentially means that if the effect on catecholamine demand is absent despite fluid resuscitation and if the fluid balance remains high, that the endothelium remains permeable and a potential positive effect of hemoadsorption is not present in these patients.

Additionally, we noted a positive correlation between lower reduction in fluid balance and higher treated blood volumes. It is crucial to highlight that factors such as the duration of hemoadsorption treatment and the maximum running rate of the adsorbers were consistent across the patient cohort. The treatment duration remained within a narrow range, between 18.43 and 20.0 h, and the maximum running rate showed minimal variation, ranging from 119.8 to 136.9 mL/min. This consistency in treatment parameters implies that differences in ABP are likely based on differences in numbers of treatments (adsorbers used), meaning that, in some cases, a more extended treatment attempt might have been made, possibly including the use of additional adsorbers. Schultz et al. found that with the application of hemoadsorption in septic patients, the observed mortality linearly decreased with blood purification volumes exceeding 6 l/kg BW [22]. These results suggest that hemoadsorption might improve survival provided that the applied dose is high enough. These findings remain in line with our data, where survivors were treated with 9 L/kg and non-survivors with 6 L/kg. Importantly, creatinine values at T_0_ were unaffected by CRRT and hemoadsorption therapy, as both procedures were started simultaneously after assessment of creatinine values at T_0_. Moreover, no ultrafiltration was performed during the T_0_–T_72_ time interval, i.e., the influence of CRRT on fluid balance is negligible. There was no difference between creatinine values at T_0_ between survivors and non-survivors (*p* = 0.296).

There are no relevant data in the literature on the administration of a second catecholamine and its effect on volume requirements. One review states that early administration of catecholamines in general may influence volume overload, and is therefore preferable [30]. Even in the Surviving Sepsis Guidelines, there are no statements about this subject, and they state that there is insufficient evidence to make a recommendation on the use of restrictive versus liberal fluid strategies in the first 24 h of resuscitation in patients with sepsis and septic shock who still have signs of hypoperfusion and volume depletion after the initial resuscitation [16]. However, the SSC Guidelines recommend a multifaceted strategy in which the initial resuscitation phase is succeeded by a more tailored approach, incorporating dynamic indices to optimize fluid therapy and to guide further fluid therapy in patients with sepsis or septic shock. Dynamic parameters, as emphasized by the relevant literature, include response to a passive leg raise [31], a fluid bolus [32], and the utilization of various hemodynamic measures such as stroke volume, stroke volume variation, pulse pressure variation [33], and echocardiography [34] where feasible.

Regarding hospital mortality amongst patients who did not receive a second catecholamine, significant reductions in fluid balance data existed between survivors (mean delta of 45.4 mL/kg) and non-survivors (mean delta of 8.5 mL/kg, *p* = 0.028), suggesting that those with greater reductions in fluid balance were more likely to survive. This stands in line with Boyd and colleagues: in their vasopressin versus norepinephrine in septic shock VASST study, they found a more positive fluid balance, whether early or cumulatively over 4 days, was associated with increased mortality. Optimal survival occurred with a positive fluid balance of approximately 3 L at 12 h [35]. However, this pattern was not evident in our patients who received a second catecholamine, who tended to be even more critically ill (higher SAPS 2 scores).

Furthermore, when analyzing hospital mortality, patients who were treated with Cytosorb hemoadsorption and received a second catecholamine had significantly better chances of survival, with an odds ratio of 2.35 and a 95% confidence interval of (1.10; 5.03). The odds of surviving to hospital discharge were almost 2.5 times higher in patients who received a second catecholamine. The mechanisms behind these observations remain, of course, purely speculative and would require further and more structured investigations.

The findings of the study by Murgolo et al. [36] provide valuable insights into the impact of “subclinical” cardiac dysfunction on fluid and vasopressor administration during the early resuscitation of septic shock. The study, conducted in patients resuscitated according to the Surviving Sepsis Campaign (SSC) Guidelines, reveals that 42% of hemodynamically stable patients, without the use of inotropes, had “subclinical” cardiac dysfunction as indicated by a Transpulmonary Thermodilution (TPTD)-derived Cardiac Function Index (CFI) ≤ 4.5 min^−1^. Notably, patients with “subclinical” cardiac dysfunction received more fluids and vasopressors during early resuscitation compared with those with normal cardiac function. The study underscores the potential influence of cardiac function on fluid and vasopressor management, even in seemingly stabilized patients. The observed impact on norepinephrine requirements and systemic vascular resistances further suggests a complex interplay between cardiac function and hemodynamic support. This research aligns with our own study’s focus on the relationship between fluid balance, catecholamine administration, and outcomes in patients receiving CytoSorb therapy. The correlation between a second catecholamine and improved survival in our study prompts consideration of factors such as cardiac function, as highlighted by Murgolo et al. Among our patient cohort, dobutamine was administered to only 17 out of 124 individuals (13.7%), comprising 11 survivors and 6 non-survivors. Given the limited number of patients receiving dobutamine, comprising a small fraction of our study population, we do not consider its impact on survival, as discussed in the work by Murgolo, to be a significant factor. It is important to note that our data collection focused on the administration of catecholamines for scoring purposes and assessing the severity of vasoplegia, rather than on the primary recording of cardiac function parameters. Consequently, we are unable to furnish data pertaining to cardiac function or septic cardiomyopathy in the context of our study. While our study did not explicitly investigate cardiac function or distinguish between types of catecholamines administered, Murgolo et al.’s findings encourage a nuanced interpretation of our results. In particular, the impact of a second catecholamine on outcomes may be influenced by underlying cardiac dynamics, which merits consideration in future research and discussions on the broader implications of CytoSorb therapy.

## 5. Limitations

This study was performed as a retrospective analysis, lacking the advantages of a prospective randomized controlled trial. The influence of country- or hospital-specific treatment protocols remains a potential confounder. Additionally, since we did not directly measure endothelial/glycocalyx function, our data only provide indirect signals through parameters influenced by endothelial function, specifically in relation to volume shifts and balances. Last, but not least, the lack of any control group including similar patients without adjunctive hemoadsorption therapy limits the interpretation of the results, and it remains unclear to what extent the observed effects on fluid balance are attributable to CytoSorb alone. Despite efforts to mitigate confounding factors, another limitation in our study is represented by the potential influence of advanced age and clinical severity, as expressed by the APACHE II score at T_0_, among hospital non-survivors. However, at the onset of septic shock (T_0_), there were no significant differences between survivors and non-survivors in either the SOFA score (total = 10.4, survivors 9.5, non-survivors 10.8; *p* = 0.106) or lactate levels (all 5.6, survivors 4.8, non-survivors 5.9, *p* = 0.192). The only distinction was a slightly higher requirement for norepinephrine in non-survivors (0.8 vs. 0.68, *p* = 0.008). Considering the SOFA score as the representative measure of organ failure severity, there is no conclusive evidence that deceased patients were more severely ill at the initial septic shock presentation. While our study provides valuable insights, the low sample size should be recognized as a potential limitation that may affect the robustness and applicability of the conclusions. This acknowledgment underscores the importance of interpreting the results within the context of the study’s inherent constraints, and it highlights the need for larger-scale investigations to validate and extend our findings in a more comprehensive manner.

## 6. Conclusions

The diverse alterations in volume balance suggest that hemoadsorption therapy may not solely target cytokine removal but may also play a pivotal role in influencing glycocalyx stability. Recent literature on CytoSorb hemoadsorption demonstrates its effectiveness in reducing a wide range of toxic substances, including endothelium-damaging proteins. Our findings show an association of the use of the applied treatment regimen including CytoSorb hemoadsorption with reductions in fluid balance, as well as vasopressor need, suggesting potential effects of this therapeutic approach in regard to stabilization of endothelial integrity. To the best of our knowledge, this is the first publication addressing this aspect in such a detailed manner. Therefore, these results, derived from a large cohort of patients, provide valuable insights that may greatly improve our understanding of the multiple effects of hemoadsorption treatment in septic shock patients.

## Figures and Tables

**Figure 1 jcm-13-00294-f001:**
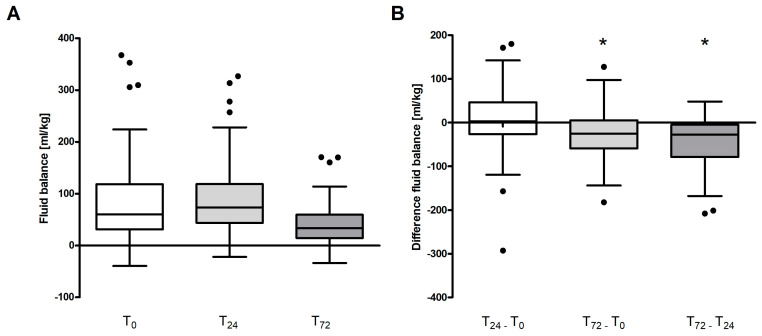
Absolute fluid balance (**A**) and calculated differences in fluid balance (**B**) for the entire study population. Depicted are Tukey boxplots with equal whisker lengths of 1.5 IQR for both whiskers. Dots represent outliers. A *p*-value of 0.05, as represented by an asterisk (*), was considered significant.

**Figure 2 jcm-13-00294-f002:**
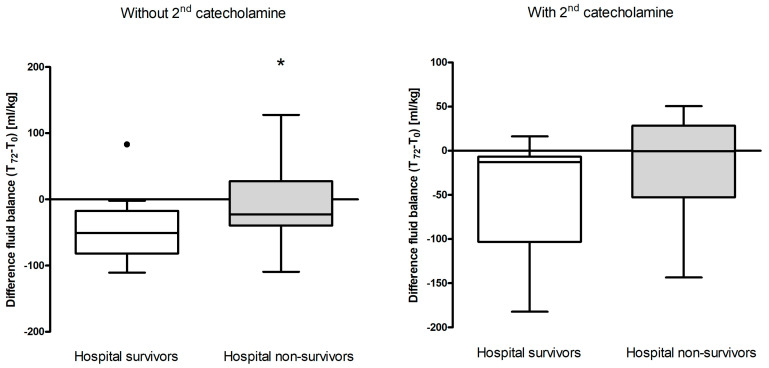
Association between fluid balance at 72 h and hospital survival in the presence of a second catecholamine. Dots represent outliers. A *p*-value of 0.05, as represented by an asterisk (*), was considered significant.

**Table 1 jcm-13-00294-t001:** Patient characteristics, treatment modalities, clinical parameters, and patient outcome.

	Overall Patient Population			Hospital Survivors			Hospital Non-Survivors			Survivors vs. Non-Survivors
	*n*	Mean	SD	*n*	Mean	SD	*n*	Mean	SD	*p*-Value
Age [years]	124	60.39	14.84	41	56.59	14.9	83	62.27	14.54	0.034
Weight [kg]	124	86.21	25.58	41	89.02	27.213	83	84.82	24.79	0.345
APACHE II Score T_0_	123	36.54	9.75	41	34.10	9.99	82	37.77	9.45	0.033
SAPS II Score T_0_	120	55.53	14.94	41	55.80	14.99	79	55.38	15.01	0.866
ICU stay [days]	124	20.56	25.40	41	30.66	19.07	83	15.57	26.72	0.001
Hospital stay [days]	124	30.13	42.36	41	48.34	34.97	83	21.13	42.98	0.001
Ventilation duration [days]	121	14.75	20.73	39	19.56	14.25	82	12.47	22.91	0.001
Therapy delay after sepsis diagnosis [h]	124	28.69	26.00	41	22.46	23.27	83	31.77	26.86	0.022
Adsorbers used [*n*]	124	2.58	1.57	41	3.37	1.69	83	2.19	1.37	0.001
DSS Score	124	7.41	1.93	41	7.39	1.76	83	7.42	2.03	0.751
ABP [L/kg]	124	6.91	0.04	41	9.00	0.05	83	6.00	0.03	0.001
SOFA Score T_0_	115	10.45	3.24	39	9.59	3.08	76	10.86	3.26	0.106
SOFA Score T_72_	91	10.53	2.92	39	9.64	2.99	52	11.19	2.72	0.025
Administered volume T_0_ [mL/kg]	115	80.43	74.62	38	76.18	73.52	77	82.53	75.56	0.649
Administered volume T_72_ [mL/kg]	91	52.36	85.57	40	36.32	32.35	51	64.94	109.52	0.031
Fluid balance T_0_ [mL/kg]	114	77.88	72.57	37	70.98	69.08	77	81.19	74.39	0.433
Fluid balance T_72_ [mL/kg]	92	40.13	39.98	40	27.11	32.83	52	50.15	42.34	0.002
Norepinephrine max T_0_ [µg/kg/min]	117	0.76	0.86	40	0.68	1.25	77	0.80	0.57	0.008
Norepinephrine max T_72_ [µg/kg/min]	92	0.47	0.51	40	0.24	0.22	52	0.63	0.59	0.001
Lactate max T_0_ [mmol/L]	109	5.64	4.19	31	4.83	4.03	78	5.96	4.24	0.192
Lactate max T_72_ [mmol/L]	85	4.08	4.96	40	1.64	1.20	45	6.25	5.95	0.001
Procalcitonin T_0_ [ng/mL]	108	30.43	47.44	41	34.76	46.97	67	27.79	47.89	0.721
Procalcitonin T_72_ [ng/mL]	82	16.66	25.91	38	20.29	31.42	44	13.53	19.83	0.395
Creatinine T_0_ [mg/dL]	110	2.78	2.02	37	3.02	2.12	73	2.65	1.97	0.296
Creatinine T_72_ [mg/dL]	88	1.48	0.81	36	1.31	0.66	52	1.59	0.89	0.191
C-reactive protein T_0_ [mg/L]	109	196.68	143.72	37	238.19	155.24	72	175.36	133.58	0.043
C-reactive protein T_72_ [mg/L]	83	168.40	126.74	35	208.74	131.20	48	138.98	116.08	0.010

**Abbreviations:** ABP—amount of blood purified, APACHE II—Acute Physiology And Chronic Health, DSS—Dynamic Scoring System, SAPS 2—Simplified Acute Physiology Score, SOFA—Sequential Organ Failure Assessment Score.

## Data Availability

The datasets used and/or analyzed during the current study are available from the corresponding author on reasonable request.

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
