# Peer review of "Impact of CytoSorb Hemoadsorption Therapy on Fluid Balance in Patients with Septic Shock"

_jcm, 2024, doi:10.3390/jcm13010294_

Round 1

Reviewer 1 Report

Comments and Suggestions for Authors

Dear Authors,

Congratulations for you paper, I have read it with great interest, since you discuss a very highly debated topic in the field of septic shock management. I have some suggestions to improve the manuscript.

MAJOR

-        You stated that your “findings showed that the use of hemoadsorption therapy results in a reduced positive fluid balance paralleled by reductions in vasopressors needs […]”. Since in this retrospective analysis there is no control group, you cannot deduce that the positive effect of reduced fluid balance and catecholamine requirement on hospital survival is attributable to Cytosorb therapy, because both groups in your study received it. This aspect is cited among limitations, but in Abstract and Conclusions section the resoluteness of your statements should be revised. Moreover, hospital non-survivors had also advanced age and clinical severity, expressed by APACHE II score at T0 and SOFA score at T72 (Table 1): this could reflect more severe disease in these patients, leading to worse outcome. Reporting data on lactate levels, norepinephrine requirements and SOFA score at septic shock onset and not only at T72 could help to compare populations and interpret results.

-        No data are reported among results on renal function and need for renal replacement therapy among included patients, although cited in Methods section. This could have a significant impact on fluid balance, even higher than hemoadsorption treatment, so it is mandatory to evaluate this parameter in your analysis.

-        In your population, the addition of a second catecholamine correlates with improved survival, and you speculate that this could be linked to positive Cytosorb effect in sicker patients. However, you did not report data on cardiac function (if evaluated) and on type of the administered second catecholamine (vasopressor or inotrope): in presence of sepsis induced cardiac dysfunction, outcome improvement may be attributable to inotrope administration more than fluid overload and vasopressors dose increase, which may worsen even more cardiac failure and significantly impact on outcome. Since septic cardiac dysfunction remains often undetected in clinical practice, it would be useful to consider this aspect in Discussion section. On this topic, you can refer to this recent paper: Murgolo, F., Mussi, R., Messina, A. et al. Subclinical cardiac dysfunction may impact on fluid and vasopressor administration during early resuscitation of septic shock. J Anesth Analg Crit Care 3, 29 (2023). https://doi.org/10.1186/s44158-023-00117-3.

MINOR

-        Introduction section (P2 L45-59): this in-depth revision of molecular mechanisms of endothelial dysfunction in sepsis is too long for this section; a shorter summary in this section with more extensive description in Discussion section could improve the readability of the paper.

-        Methods section (P3) – Study design: you did not report any exclusion criteria, please declare the absence and justify the reason.

-        Methods section (P3 L102-103): administered fluid resuscitation volumes are reported among primary endpoints, but in Results section the datum is cited only in Table 1 and it misses in sequent analysis. Please revise it.

-        Methods section: you did not report data on timely source control and appropriate antibiotic treatment start, which are two cornerstones in septic shock management, together with hemodynamic resuscitation: please consider these parameters or discuss the absence of these data in results interpretation, since these two aspect may significantly impact the outcome.

-        Results section (P4): it would be interesting to evaluate type of infection in survivors vs. non-survivors, since cytokines burden and biomarkers vary a lot when considering abdominal sepsis or urosepsis compared to pneumonia, and this could lead to different effect of Cytosorb therapy.

-        Table 1 (P4): as previously suggested, data on SOFA score, fluid volume and fluid balance, norepinephrine dose, lactate levels and inflammatory biomarkers at T0 should be reported. Moreover, fluid volume should be reported as ml/kg more than total litres, to improve result interpretation. Hydrocortisone administration should be reported per group (not only in overall population as reported in text) and added in Table 1. Data on ICU length of stay, renal function and renal replacement therapy are missing, even if cited in Methods section.

Reviewer 2 Report

Comments and Suggestions for Authors

I read with great interest the manuscript by Kogelmann et al. on the role of Cytosorb on fluid balance in septic shock patients. The study is sound and well written. However, there are some issues that need to be addressed:

- Line 59. Before discussing the role of Cytosorb in sepsis, authors should report that surviving sepsis campaign guidelines (https://doi.org/10.1007/s00134-021-06506-y) state that to date there is insufficient evidence to make a recommendation on the use of blood purification techniques. For this reason, further studies are needed to explore the role of blood purification strategies.

- Was the informed consent waived for these patients? Please specify.

- Line 124. Beside the first hours, was hemodynamic monitoring used in these patients for the optimization phase?  Please specify.

- Line 254-258. Authors should explain the recommendations of the SSC more in detail. In fact, guidelines suggest an initial phase of fluid resuscitation followed by the use of dynamic indices to guide fluid therapy in patients with sepsis or septic shock. Dynamic parameters include response to a passive leg raise (doi: 10.1007/s00134-015-4134-1) or a fluid bolus (https://doi.org/10.1016/j.tacc.2023.101316), using stroke volume, stroke volume variation, pulse pressure variation (doi: 10.1097/EJA.0b013e32834b7d82), or echocardiography (doi: 10.1186/s40635-023-00529-z), where available. Please discuss and add these 4 references.

- Authors should report the low sample size as a possible further limitation of the study.

Round 2

Reviewer 1 Report

Comments and Suggestions for Authors

Dear Authors,

Thanks for revising your manuscript. I have no other suggestions.